# OpenReview forum: "Automatically Finding Reward Model Biases"
_ICLR.cc/2026/Workshop/AFAA — AFAA 2026 Poster_

### Official Review · Reviewer_rHyZ · 2026-02-18
**Automatically Finding Reward Model Biases**

**Rating:** 4
**Confidence:** 4

**Summary:**

This paper proposes a **black-box auditing method to automatically discover reward model (RM) biases**—situations where an RM systematically assigns higher reward to response attributes that a competent judge would consider undesirable (e.g., artifacts, hallucinated evidence, superficial formatting hacks). The method generates candidate “bias attributes,” validates them with counterfactual pairs (attribute present vs absent) created via minimal rewrites, and uses an iterative search procedure to amplify and diversify the most problematic biases (high RM preference but low judge preference). The goal is to catch reward-model quirks early, before running expensive RLHF that could exploit them.

**Strengths:**

* **Black-box and broadly applicable in principle**: The approach only needs query access to the RM plus a judge/rewriter, so it can be used even when RM internals, training data, or gradients are unavailable.

* **Finds concrete and surprising failure modes**: The surfaced biases are not just abstract (e.g., formatting artifacts like redundant spacing, or rewarding fabricated “supporting evidence”), which strengthens the case that the method can uncover non-obvious vulnerabilities.

**Weaknesses:**

* **High dependence on the initial seed / starting set**: The discovered biases can be strongly influenced by the initial candidate pool; if the seed set misses important regions of the space, the iterative procedure may never find certain bias families.
* **LLM-as-a-Judge (LLMaaJ) introduces its own biases**: Even frontier judges can have preference artifacts; relying on a single judge risks baking judge-specific quirks into what is labeled “undesirable.” Using multiple judges (or ensembling) and reporting agreement could reduce this concern.
* **LLM rewriter confounds (unintended attribute changes)**: The “minimal rewrite” counterfactual construction is still vulnerable to correlated edits—rewriting to add/remove one attribute can unintentionally change other attributes (tone, specificity, factuality), weakening causal attribution.
* **Limited RM coverage**: Demonstrating the method on a broader set of reward model families (and/or larger, higher-capacity RMs) would strengthen the claim that this is a general auditing tool rather than a case study on one RM. Comparing how discovered attributes differ across RM families could provide additional insight into what each RM is over-optimizing.
* **Limited insight into attribute evolution**: The paper would benefit from deeper analysis of *how* attributes mutate/refine across iterations (e.g., clustering discovered attributes, showing trajectories, or summarizing common “routes” to higher RM preference), which would help interpret the search dynamics and guide mitigation.

---

### Official Review · Reviewer_ZnWe · 2026-02-21
**Automated Discovery of Reward Model Biases: Relevant and Timely, with Limited Scope**

**Rating:** 4
**Confidence:** 3

**Summary:**

The paper introduces the problem of automatically identifying biases in reward models (RMs) used in LLM post-training. The authors propose an iterative method in which an LLM generates and refines candidate bias hypotheses, which are then evaluated against the reward model’s scoring behavior. The approach is shown to recover known biases and surface novel ones. The paper also demonstrates that iterative refinement outperforms flat best-of-N search in identifying such biases.

**Strengths:**

1. Addresses an important and underexplored problem: automated detection of reward model biases.
2. Simple and practical method that leverages LLM-based iteration rather than requiring heavy supervision.
3. Successfully recovers known biases while identifying new ones, demonstrating practical utility.
4. Strong relevance to the workshop’s focus on fairness in alignment procedures and reward modeling.

**Weaknesses:**

1. The empirical evaluation focuses primarily on a single open-weight reward model, limiting evidence of generalizability.
2. Bias validation is primarily based on reward model scoring behavior rather than external human evaluation.
3. The paper adopts a broad behavioral notion of bias without connecting it to formal fairness frameworks.
4. The work focuses on bias detection and does not empirically demonstrate mitigation or downstream alignment improvements.

---

### Official Review · Reviewer_qzRF · 2026-02-23
**Automated RM-bias auditing framework with strong exploratory evidence, but limited breadth and underpowered comparative validation.**

**Rating:** 3
**Confidence:** 4

**Summary:**

This paper studies automatic discovery of reward-model biases and proposes an iterative evolutionary pipeline that generates, mutates, and filters natural-language bias hypotheses. The method defines a bias as an attribute preferred by the target reward model but dispreferred by a stronger LLM judge, estimated using counterfactual rewrites and two-objective filtering (Section 2.1-2.3). On Skywork-Reward-V2-Llama-3.1-8B, the pipeline produces 17 candidates and reports 10 statistically significant biases on a held-out test set (Table 1, Appendix F.1). The paper includes two case studies (triple-space formatting and hallucinated-details style behavior), rewrite-fidelity and invariance analyses, and a depth-vs-width configuration comparison for the search pipeline (Section 3.3). The contribution is positioned as a black-box, interpretable RM-auditing method that can surface both known and previously under-discussed failure patterns.

**Strengths:**

- Important problem setting with practical value: black-box RM auditing before policy optimization is useful and timely (Section 1, Section 2.3).
- Clear formalization of bias as disagreement between RM preference and judge preference over counterfactual pairs (Section 2.1), with explicit two-objective optimization and Pareto filtering (Algorithm 1, Appendix C.3).
- Solid engineering/validation effort for rewrite quality: fidelity results (Table 3), multi-rewriter agreement/invariance checks (Appendix D.2), and rubric-based quality inspection (Appendix D.3).
- Statistical filtering is stronger than many exploratory audits: one-sided tests with multiple-testing correction, plus partial conjunction testing for cross-rewriter significance (Section 3.2, Appendix F.1).
- Interesting findings beyond standard length/format concerns, including the political-topic triple-space artifact and made-up-event hallucinated detail tendency (Section 3.2, Table 2, Table 5).
- The draft is generally clear and coherent; problem setup, algorithm flow, and case-study narrative are easy to follow.
- Even with limited breadth, this is a useful exploratory artifact: practical pipeline, statistically filtered candidates, and concrete failure-mode examples that could inform follow-up robustness/debiasing work.

**Weaknesses:**

- Limited external validity: core conclusions are from one RM family (Skywork-V2-8B) and synthetic topic-conditioned prompts (Section 2.2, Section 5), so generalization beyond this setup is uncertain.
- "Undesirable to humans" remains proxy-heavy and judge-dependent. The same LLM judge is used in the formal definition of bias (\(J(A) < 0.5\), Section 2.1), iterative candidate selection, and final validation/test significance criteria (Section 2.3, Section 3.2). The paper includes manual final screening and qualitative examples, but no systematic human study (annotator protocol, blinded labeling, per-attribute agreement, inter-rater reliability). This is important for subtle/normative attributes (e.g., B5/B8/B12 in Table 1), where judge-specific preferences may affect significance outcomes.
- Comparison to adjacent prior work is not sufficiently direct. The paper argues prior methods primarily find instance-level adversarial examples and not natural-language bias descriptions, but this is not directly validated with matched-budget experiments. It compares only internal ablations (depth-vs-width, Section 3.3), not head-to-head baselines against close failure-discovery methods such as REFORM or Adv-RM. These methods have adjacent but not identical objectives; still, without direct comparison it is difficult to isolate gains in discovery rate, diversity/coverage, interpretability, or compute cost per validated bias.
- Pipeline-configuration claim is underpowered: depth-vs-width is run once per setting and the paper explicitly notes conclusions are not rigorous (Section 3.3). Because the search is stochastic (LLM hypothesis generation, mutation, rewrite-based scoring), single runs cannot separate true configuration effects from seed/topic noise or estimate variance. The triple-space case study also shows artifact sensitivity, where ancestry traces back to a typo in baseline samples (Section 3.2, Table 2).
- No downstream impact test: unlike adjacent robustness-oriented works that evaluate BoN/RLHF-type outcomes, this paper does not test whether removing discovered biases improves policy behavior.
- Writing quality is generally good, but a few copy-editing issues remain (e.g., “Anthopic,” “judge threshold is swept,” and awkward phrasing such as “still worthy of manual inspection”).

---

### Meta-Review · Area_Chair_VMY7 · 2026-02-24

**Recommendation:** Main Papers Track
**Confidence:** 4

**Metareview:**

The paper tackles a timely topic and provides a black-box and broadly applicable approach. Reviewers agree that it's relevant to the workshop and that what was proposed is  Reviewers agree that the evaluation has several blind spots and could require improvements (e.g., comparison with prior work, bias validation being based on reward model scoring behavior, reliance on LLM-as-a-judge). However, the paper has its merit and could be the start of interesting subsequent work and discussions.

---

### Decision · Program_Chairs · 2026-03-02

Accept (Poster)